

# Structural analysis of S-wave seismics around an urban sinkhole; evidence of enhanced subrosion in a strike-slip fault zone

*Sonja H. Wadas*[1], *David C. Tanner*[1], *Ulrich Polom*[1], and *Charlotte M. Krawczyk*[2,3]

[1]Leibniz Institute for Applied Geophysics, Stilleweg 2, D-30655 Hannover, Germany
[2]GFZ German Research Centre for Geosciences, Telegrafenberg, D-14473 Potsdam, Germany
[3]Technical University Berlin, Ernst-Reuter-Platz 1, D-10587, Germany

*Correspondence to:* Sonja H. Wadas (sonja.wadas@liag-hannover.de)

**Abstract.** In November 2010, a large sinkhole opened up in the urban area of Schmalkalden, Germany. To determine the key factors which benefited the development of this collapse structure and therefore the subrosion, we carried out several shear wave reflection seismic profiles around the sinkhole. In the seismic sections we see evidence of the Mesozoic tectonic movement, in the form of a NW–SE striking, dextral strike-slip fault, known as the Heßleser Fault, which faulted and fractured the subsurface

below the town. The strike-slip faulting created a zone of small blocks (<100 m in size), around which steeply-dipping normal faults, reverse faults, and a dense fracture network serve as fluid pathways for the artesian-confined groundwater. The faults also acted as barriers for horizontal groundwater flow perpendicular to the fault planes. Instead the groundwater is flowing along the faults which serve as conduits. A near-surface fault zone located in soluble rocks can enhance subrosion in two ways: (1) tectonic movements accompanied by strain variations lead to the formation of small fault blocks and a fracture network that

increases the rock permeability and creates fluid pathways, and (2) the faults can serve as groundwater conduits. Also note, the more complex the fault geometry and the more interaction between faults, the more fractures are generated and the more prone to sinkholes occurrence is the area.

## 1  Introduction

Sinkholes are caused by 'subrosion'. The term 'subrosion' describes the underground leaching of soluble rocks such as salt,

sulfate and carbonate in the presence of water (e.g., groundwater). Fractures and faults can serve as fluid pathways which allow the water to flow through the subsurface and thus generate cavities. A number of authors have pointed out that there is a causal connection between strike-slip faults and sinkholes (Heubeck et al., 2004; Closson & Karaki, 2009; Gabbianelli et al., 2009; Lunina et al., 2016). Strike-slip fault zones, with different internal types of fault sense (reverse, normal and strike-slip), are most-likely to produce a fine mosaic of small fault blocks that allow groundwater to move freely, thus creating an area

of subrosion. Over time, different subrosion structures can occur depending on, e.g. the solubility of the rocks, the flow rate, and the type of the overburden (e.g., soft sediments or solid rock). The two main subrosion features that can evolve close to the surface are collapse or depression structures. The former occurs if the overburden is thin enough, the latter is due to slow dissolution (Smyth, 1913; White & White, 1969; Beck, 1988; Martinez et al., 1998; Yechieli et al., 2002; Waltham et al.,



2005; Gutiérrez et al., 2008). Sinkholes can cause damage to buildings and infrastructure, and may even lead to life-threatening situations, if they occur e.g. in urban areas.

Subrosion is a barely-understood phenomenon. To determine the causes and the main controlling factors of the sinkhole formation in the urban area of Schmalkalden, a number of investigations were conducted on behalf of the Thuringian State

Institute for Environment and Geology (TLUG), including investigation of possible man-made underground cavities, boreholes, micro-gravimetry, 2D compression wave (P-wave) reflection seismic, and hydrological investigations (e.g., chemical composition of the four aquifers) (Schmidt et al., 2013).

The P-wave reflection seismic was unsuccessful to image the first ca. 30 m below surface due to a relatively poor resolution, but for instance shear waves (S-wave) are able to image the near-surface in high resolution (Dobecki & Upchurch, 2006;

Krawczyk et al., 2012; Polom et al., 2016a; Wadas et al., 2016). Interpretation of near-surface faults and structures from the surface down to ca. 100 m depth is important for understanding the local geology and the subrosion structures and processes in general. Therefore Leibniz Institute for Applied Geophysics (LIAG) carried out 2D $S_H$-wave reflection seismics in this area.

## 2 Study area

### 2.1 Geological evolution

Schmalkalden is located in southern Thuringia in Germany. The deeper bedrock below the research area consists of metamorphic gneiss and micaceous shale, which were deformed during the Variscan Orogeny. Due to erosion of the overburden during the Upper Carboniferous and the Lower Permian, WNW–ESE and NE–SW striking valleys were formed (Wunderlich, 1995). Later this bedrock was uplifted and formed the Ruhla-Schmalkalden Horst (RSH). The metamorphic bedrock does not contain soluble rocks that could be affected by subrosion.

During the Upper Permian, the Zechstein Sea transgressed, but due to the horst location, the sediment strata are much thinner than elsewhere in the basin. The climate conditions of the Permian lead to deposition of large thicknesses of evaporite. The Zechstein deposits are subdivided into seven sequences, which begin with reef dolomite rocks, which surround the RSH and form the Werra Formation. The upper part of the Werra Formation is characterized by red claystones which indicate the end of the reef growth (Dittrich, 1966). Sulfate rocks occur above this, which form the main subrosion horizon. The following Staßfurt

Formation consists of sulfates, claystones and dolomites, whereby over 50% of the sulfate rocks are gypsum, which is why this formation is categorized as part of the subrosion horizon. This formation is followed by the Leine Formation, which contains claystones and carbonates. The upper part of the Zechstein deposits is represented by claystones, sandstones, and dolomites of the Leine-, Aller-, Ohre- and Friesland Formations and it finishes with sand- and claystones of the Fulda Formation (Schmidt et al., 2012, 2013).

In the study area the Zechstein Formation is followed by terrestrial sediments of the Triassic; the Calvörde and the Bernburg Formations of the Lower Buntsandstein (Fig. 1). Because of intense erosion due to fault movement, mostly since the Upper Cretaceous, which also lead to the uplift of the Thuringian Forest, these formations are also the youngest in the region, except for some Quaternary deposits.



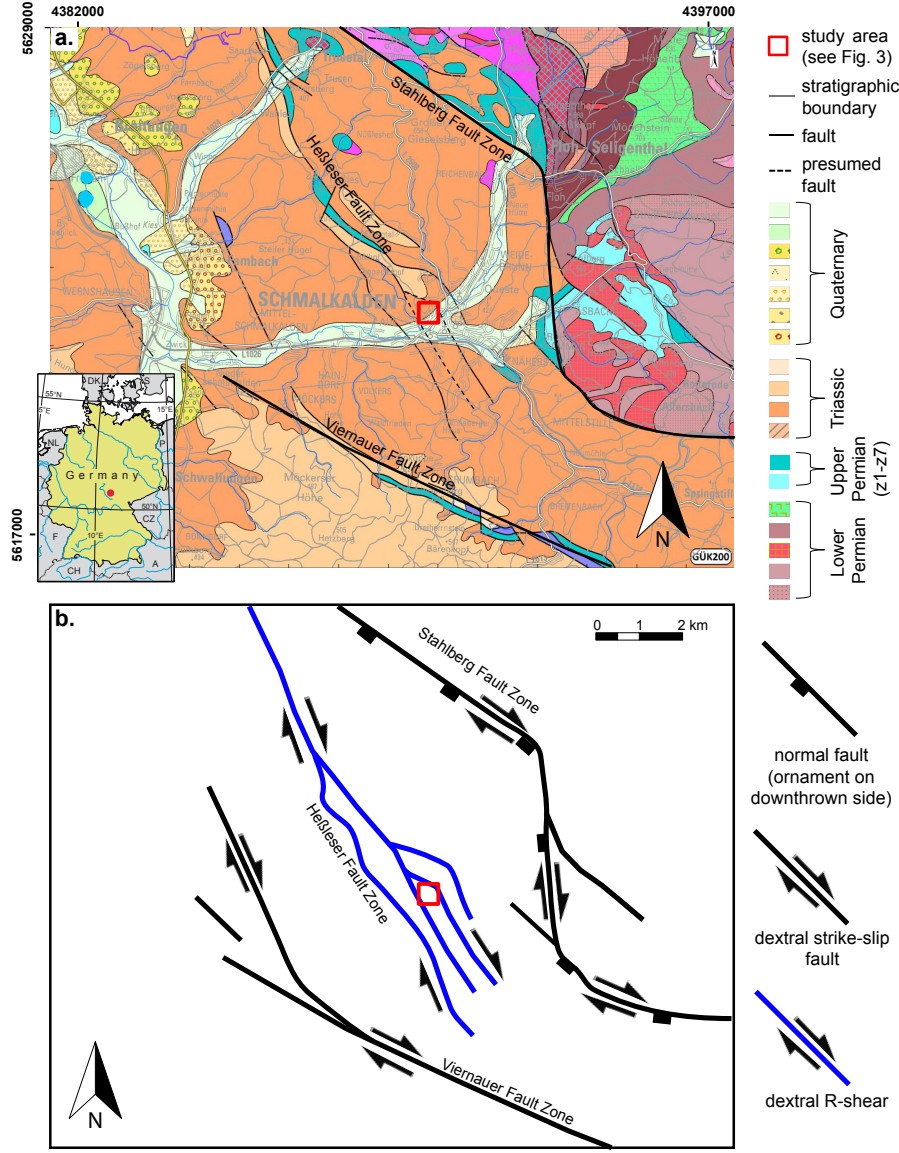

**Figure 1.** Geological map (a) of Schmalkalden and the surrounding areas (after Bücking, 1906) showing the Heßleser Fault Zone (HFZ) crossing the position of the sinkhole (red square). In the lower left corner a map of Germany with the position of Schmalkalden is shown (red dot). The Mesozoic movements reactivated the tectonic faults (b) and lead to the formation of a dextral strike-slip fault zone which includes the Stahlberg Fault Zone (SFZ) and the Viernauer Fault Zone (VFZ). The HFZ connects the two major faults at an acute angle of 30°. We interpret this as a Riedel R-shear also dextral in movement (see text).

Altogether, six tectonic phases between the Lower Carboniferous and the Tertiary have been distinguished, of which the last two are the most important for this work. From the Upper Permian to the Lower Cretaceous the area of Schmalkalden





was subject to an extensional stress regime and from the Upper Cretaceous to the Tertiary the area was dominated by a compressional stress regime (Wunderlich, 1997b).

## 2.2 Faults

Thuringia is crossed by several major NW–SE striking faults (Wunderlichet al., 1997a; Wunderlich, 1997b; Andreas & Wun-
derlich, 1998). Schmalkalden is located to the south of the Stahlberg Fault Zone (SFZ) in this area. The SFZ is downthrown to the southwest and raises basement rocks to the northeast. However, this was accompanied by dextral strike-slip movement, as can be seen from the jogs in the fault trace (Fig. 1). Together with the Viernauer Fault (VFZ) to the south, the SFZ and the VFZ formed a dextral strike-slip fault zone. The Heßleser Fault Zone (HFZ) connects the two major faults at an acute angle of 30°. We interpret this as a Riedel R-shear (for definition of a Riedel shear, see Woodcock & Fischer (1986)), also dextral
in movement (Fig. 1). Most probably this movement took place during the Upper Cretaceous/Early Tertiary inversion phase in Europe (Tanner et al., 1998; Littke et al., 2008; Kley & Voigt, 2008; Tanner & Krawczyk, 2017). The southeastern part of the HFZ crosscuts the town of Schmalkalden. The fault zone contains several smaller fault branches that strike NW–SE (Bücking, 1906; Böhne, 1915; Krzywicki, 1937).

## 2.3 Sinkhole

On 1st November 2010, at half past three in the morning, a large sinkhole opened up in the residential area of Schmalkalden (Fig. 2). The sinkhole was 26 m to 30 m in diameter, 12 m to 17 m in depth and the crater had a volume of 4000 m$^3$ to 4200 m$^3$. Directly after the collapse, groundwater fountains were observed along the sinkhole margins and the sinkhole filled with water, which subsequently slowly seeped away into the ground. The crater became wider due to instability of the sinkhole margins. The bedrock, which was visible within the crater, was strongly fractured and showed small-scale faults and folding of layers.
The damage caused by the collapse, such as cracks in houses and streets, was mainly concentrated on the areas north and northeast of the sinkhole. The houses directly besides the sinkhole were temporarily closed for safety reasons. New cracks formed due to slope movements along the south-west dipping layers, caused by the reduced bedrock stability. To stop slope movement the sinkhole was quickly filled with gravel.

## 2.4 Stratigraphy

To investigate the stratigraphy in the vicinity of the sinkhole in detail, five boreholes with depths of 143 m to 167 m were drilled (Fig. 3). In the following, the stratigraphy of borehole 05/2011 is shown as an example (TLUG, 2017a). The first 3.55 m consist of anthropogenic deposits, and from 3.55 m to 13 m depth, Quaternary terrace gravels and sandy colluvial deposits are found. Between 13 m and 28 m depth are red sandstones of the Triassic Buntsandstein (Calvörde Formation) followed by deposits of the Permian Zechstein. The Zechstein is subdivided into seven formations (see previous section) of which the Fulda
Formation (z7) from 28 m to 43.45 m depth is the youngest, which consists of silty sandstones and claystones (also called 'Bröckelschiefer'). Below z7, from 43.45 m to 58.60 m, the Aller-, Ohre- and Friesland Formations (z4–z6) and the upper





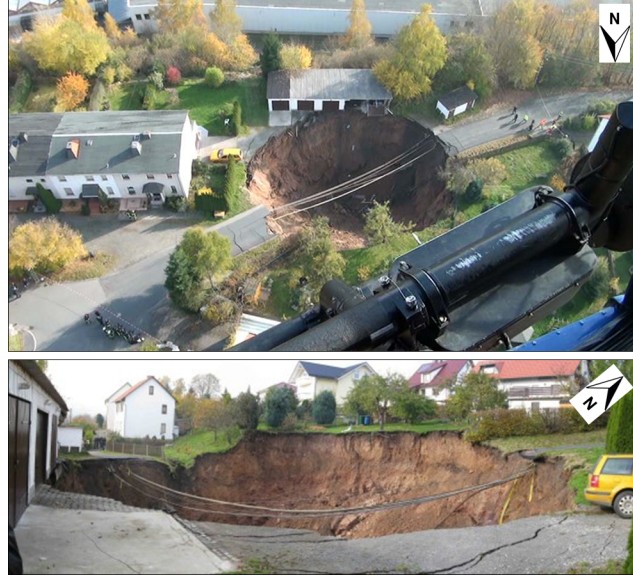

**Figure 2.** Photographs showing the sinkhole in Schmalkalden that opened up on 1st November 2010. It is 26 to 30 m in diameter and 12 to 17 m in depth (TLUG, 2010).

Leine Formation (z3Tb) are found with disturbed clay- and sandstones. Between 58.60 m and 86.65 m are disturbed dolomitic lime- and claystones of the lower Leine Formation (z3Ca & z3Ta). The Staßfurt Formation (z2) from 86.65 m to 106.85 m depth consists of claystone and gypsum. The oldest Zechstein sequence, the Werra Formation (z1), is found between 106.85 m and 150.00 m, and consists of a gypsum and anhydrite subrosion breccia, claystone and carbonate. The top of the bedrock, the
Paleozoic Hohleborn Formation, was drilled by the other four boreholes (Fig. 3).

Important results were the discovery of faults and the verification that the subrosion horizon is situated between the base of the Leine Carbonate and the base of the Werra Anhydrite.

## 3   Shear wave data

Seven $S_H$-wave reflection seismic profiles (total length of ca. 1.3 km) were acquired in the vicinity of the sinkhole in March
2016 and February 2017 in two field campaigns (Fig. 4). To generate horizontally-polarized S-waves the micro-vibrator ELVIS 7 (Polom, 2003; Druivenga et al., 2011) was used (Fig. 5). The sweep had a duration of 10 seconds over a frequency range of 20 Hz to 120 Hz and the source spacing was 2 m and 4 m. A receiver array of 112/120 one-component horizontal geophones at 1 m spacing and attached to a landstreamer was used and Geometrics Geodes recorded 12 seconds of raw signal (Fig. 5 & Tab. A1). A variable split-spread geometry was used with the source and the receivers moving forward. After surveying
60 m the receivers were moved 60 m forward, while the source moved 2 m or 4 m forward continuously. The survey setup is designed for near-surface reflection seismic profiling especially in urban areas, e.g. the suppression of surface Love waves, if





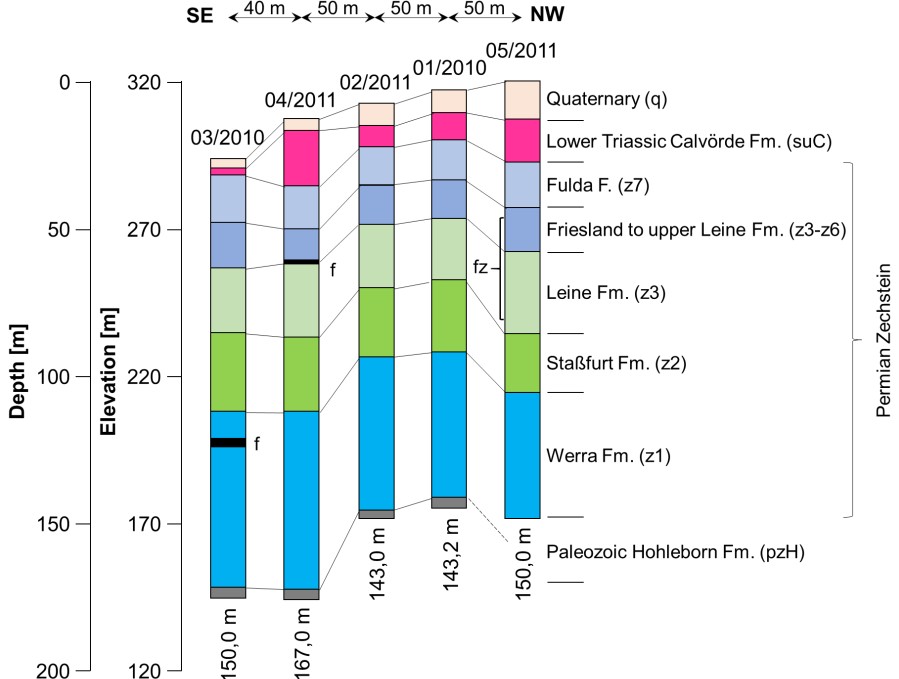

**Figure 3.** Stratigraphy of the five boreholes (TLUG, 2017a) near the sinkhole (f = fault, fz = fault zone). The numbers below the profiles show the final depth below surface of each borehole. The stratigraphic units were used for the interpretation of the seismic profiles.

the first subsurface layer is of higher seismic velocity than the second layer, which is often the case on paved or compacted roads (Polom et al., 2010; Krawczyk et al., 2012). Important for a successful survey in a subrosion area are the use of a high frequency signal, and a dense receiver and source spacing, in order to detect the strong lateral and vertical, structural variations induced by subrosion and to get high-resolution images of the subsurface (Wadas et al., 2016). For more information see

5    Krawczyk et al. (2013); Polom et al. (2013).

$S_H$-wave reflection seismic data processing was carried out using the VISTA software (version 10.028.1895). During pre-processing each record was visually examined for quality assessment, then geometry istallation and vibroseis-correlation were carried out. This was followed by a 2-fold vertical stack, automatic gain control, normalization and bandpass filter. Afterwards top mute and spectral balancing were applied, which is important for profiles showing a high frequency attenuation. Surface

10   waves and harmonic distortions covering the desired signal were removed by a frequency-wavenumber (FK) filter. An interactive velocity analysis, normal moveout correction and CMP stack were carried out to generate a seismic section in time domain. Finally, finite-difference (FD) time migration and time-to-depth conversion were applied (Tab. A2).





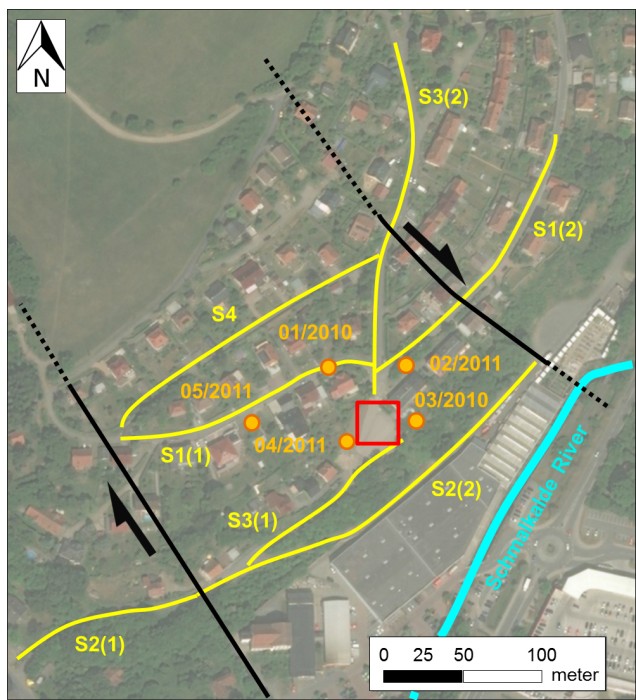

**Figure 4.** Location map of the $S_H$-wave reflection seismic profiles (yellow lines), the boreholes (orange dots), the sinkhole (red square) and the main fault branches of the Heßleser Fault Zone (black lines) (ArcGIS, Open Source Map).

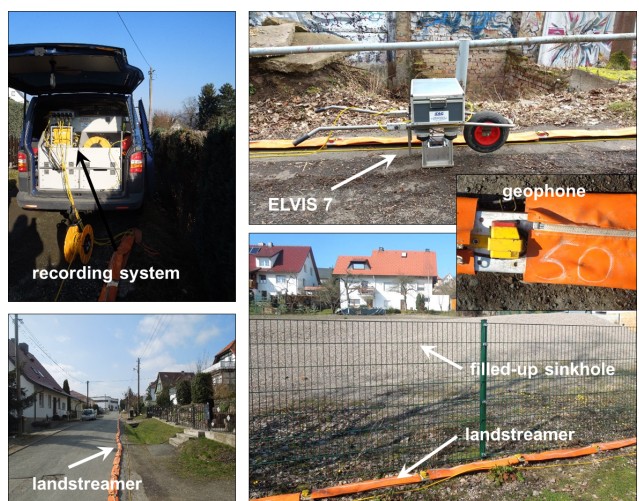

**Figure 5.** Photographs taken during the two field campaigns showing the micro-vibrator, the landstreamer and the recording system used for the $S_H$-wave reflection seismic surveys.



## 4  Results

### 4.1  Seismic interpretation of S1

$S_H$-wave reflection seismic profile S1 of ca. 350 m length, was carried out north of the sinkhole (Fig. 4). At ca. 10 m to 25 m depth, a continuous reflector with high amplitudes is traceable throughout the entire profile (Fig. 6). The strong impedance

5  contrast represents the boundary between the Triassic sandstones of the Calvörde Formation (suC) and the Permian claystones of the Fulda Formation (z7). This reflector, which can be found in all $S_H$-wave reflection seismic profiles, was used as a marker horizon. In contrast, the area beneath shows a mostly discontinuous reflection pattern with no remarkable reflector, but instead, lateral amplitude variations are observed due to strongly fractured strata within the seven Zechstein Formations.

In the area northwest of the sinkhole, shallowly-dipping reflectors form a bowl-shaped structure of ca. 150 m length and

10  50 m depth within the Triassic and Permian deposits. In this area the Calvörde Formation shows local thickening. Numerous faults in the Zechstein Formations with small-scale vertical offsets of ca. 1 m to 3 m were imaged.

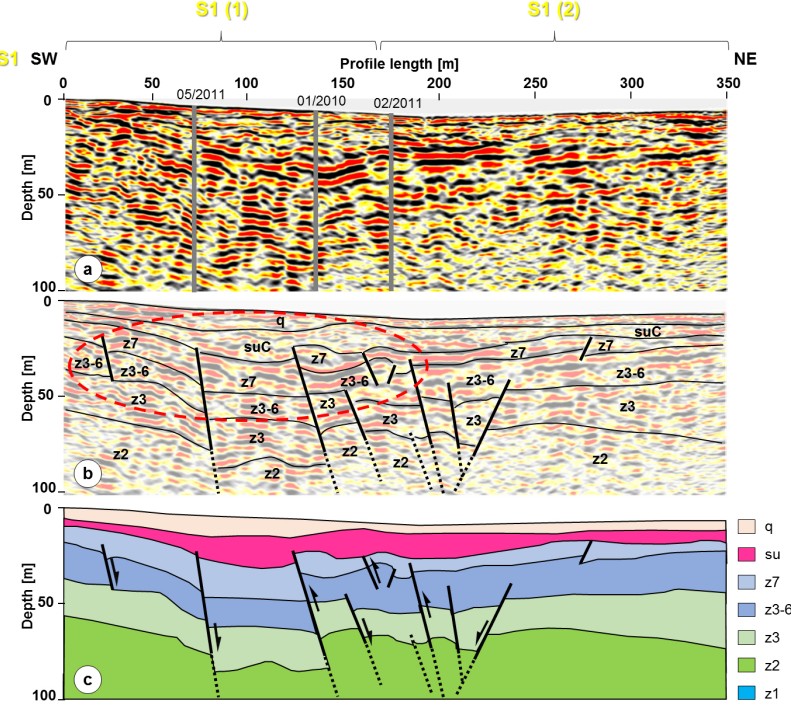

**Figure 6.** Reflection seismic profile S1 uninterpreted (a), with stratigraphy derived from boreholes 05/2011, 01/2010 and 02/2011 (b) and with interpretation (c). The stratigraphic units are explained in Fig.4. The profile was surveyed north of the sinkhole. Besides local thickening of sediments and thinning of evaporites, several steep-dipping normal and reverse faults within the Permian and Triassic deposits were identified. In the southwest a bowl-shaped structure (red circle) is visible, which was interpreted as a subrosion-induced depression.





A low reflectivity zone (LRZ) can be observed at ca. 70 m to 100 m depth between 175 m and 275 m profile length. This zone shows an almost transparent reflection pattern compared to the neighbouring reflectors at the same depth and is located within the Staßfurt Formation. This correlates with the subrosion horizon within the Zechstein formations of Werra Anhydrite to the Leine Carbonate (z1-z3).

5    The near-surface in all profiles, is imaged in detail, with a resolution of less than 1 m at depths down to ca. 15 m and a resolution of ca. 1 m to 3 m at 50 m depth.

### 4.2 Seismic interpretation of S2

$S_H$-wave reflection seismic profile S2 of ca. 400 m length, was carried out south of the sinkhole (Fig. 4). The marker horizon which represents the base of the Calvörde Formation is clearly visible at ca. 10 m to 15 m depth and is traceable throughout

10   the entire profile (Fig. 7). The Permian deposits below show the same discontinuous and disrupted pattern as in profile S1 and numerous fractures were imaged.

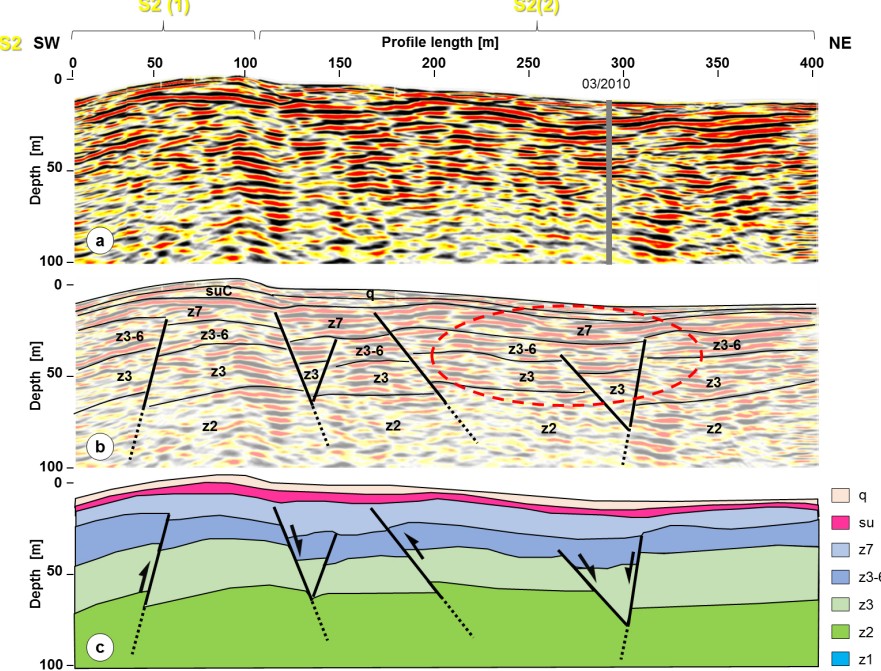

**Figure 7.** Reflection seismic profile S2 uninterpreted (a), with stratigraphy derived from borehole 03/2010 (b) and with interpretation (c). The stratigraphic units are explained in Fig.4. The profile was surveyed south of the sinkhole. Steep-dipping normal and reverse faults within the Permian deposits were identified and in the northeast a bowl-shaped structure (red circle) is visible, which was interpreted as a subrosion-induced depression.



In the northeastern area, dipping reflectors form a bowl-shaped structure of ca. 100 m length and 40 m depth within the Triassic and Permian deposits. Below this depression a steep normal fault dipping to the southwest was identified by the reflection seismic profile and the borehole 03/2010. In the southwest of the seismic section other normal and reverse faults with vertical offsets of ca. 5 m to 10 m were imaged.

5    In the Staßfurt Formation, which includes to the subrosion horizon, a large LRZ is observed between 175 m and 300 m profile length at ca. 70 m to 100 m depth below the depression.

### 4.3    Seismic interpretation of S3

$S_H$-wave reflection seismic profile S3 of ca. 370 m length (including a 30 m gap around the sinkhole), was carried out from south to north, passing the sinkhole (Fig. 4). The reflection pattern is similar to S1 and S2.

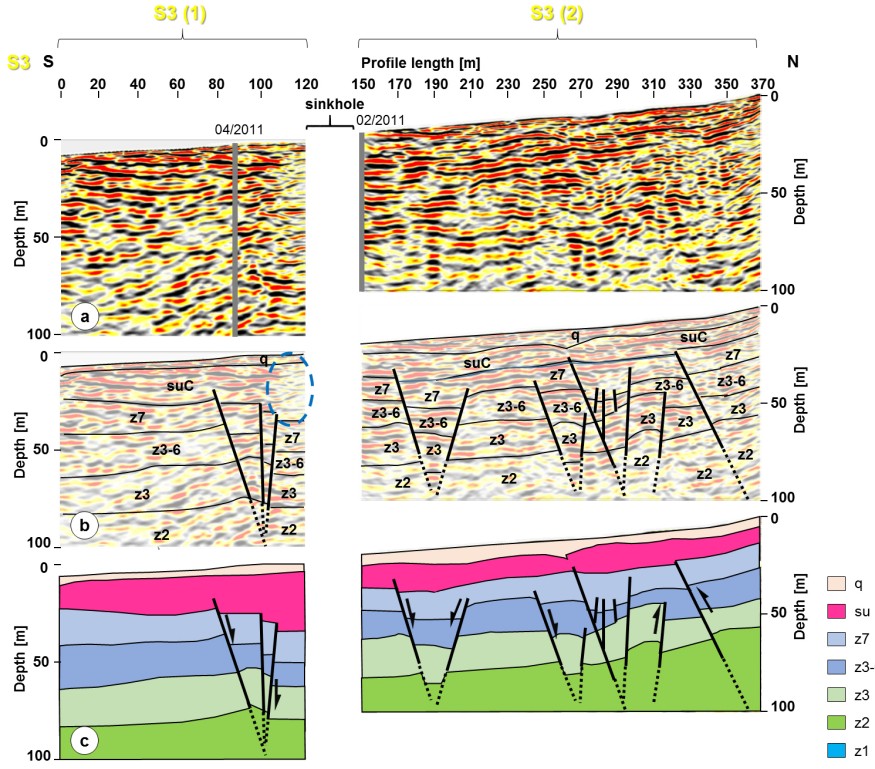

**Figure 8.** Reflection seismic profile S3 uninterpreted (a), with stratigraphy derived from boreholes 04/2011 and 02/2011 (b) and with interpretation (c). The stratigraphic units are explained in Fig.4. The profile was surveyed from south to north across the sinkhole area leaving out the sinkhole itself. Just like in profiles S1 and S2 steep normal and reverse faults can be seen, but no bowl-shaped structure. Instead the southern sinkhole margin is visible as a transparent area (blue circle) with near-surface reflectors dipping to the direction of the sinkhole.





The flat-lying, mostly continuous reflectors of the Quaternary and the marker horizon of the Triassic Buntsandstein can be precisely identified (Figs. 8). The reflection pattern of the Permian is discontinuous due to vertical displacements of reflectors. Several near-surface normal and reverse faults were identified with fault offsets of ca. 10 m to 20 m.

Bowl-shaped structures, as seen in S1 and S2, are not imaged, but between ca. 100 m and 120 m profile length, an almost

5    transparent area at ca. 40 m depth can be observed. This part of S3 was surveyed alongside the filled sinkhole margin. No layers can be identified within the sinkhole zone; the reflectors south of it dip towards the direction of the sinkhole. South of this structure, between 80 m and 120 m profile length, a northward-dipping steep normal fault was identified, which was drilled by borehole 04/2011.

Small LRZ's are observed, e.g. between 240 m and 270 m profile length at 70 m to 95 m depth.

10   ## 4.4   Seismic interpretation of S4

$S_H$-wave reflection seismic profile S4 of ca. 190 m length, was carried out northwest of the sinkhole (Fig. 4). The Quaternary and the Triassic deposits were identified using the marker horizons and the stratigraphy of borehole 01/2010, which was projected onto the seismic line (Fig. 9). In the northeast and the southwest, within the discontinuous and displaced Zechstein

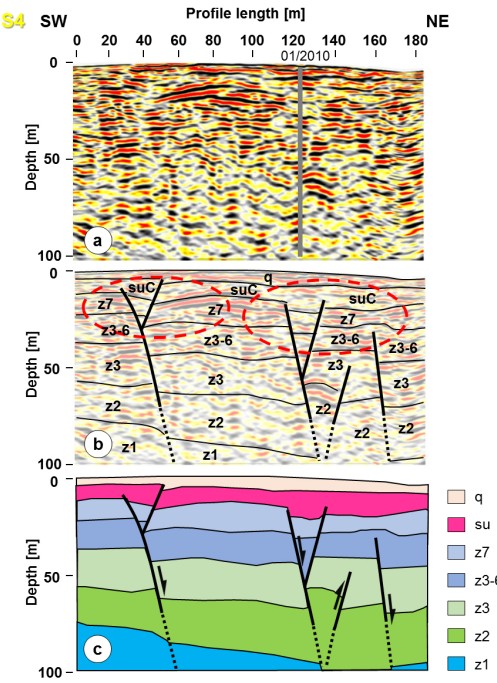

**Figure 9.** Reflection seismic profile S4 uninterpreted (a), with stratigraphy derived from borehole 01/2010 (b) and with interpretation (c). The stratigraphic units are explained in Fig.4. The profile was surveyed north of the sinkhole parallel to S1(1). The normal faults seen in S1 were also identified in S4 and two almost bowl-shaped structures (red circles) are visible in the southwest and the northeast.





Formations, faults with vertical offsets of 5 m to 10 m were imaged, which are probably the same as those seen in S1, since S4 runs parallel to the western part of S1. In the same areas two almost bowl-shaped structures can be identified down to ca. 20 m depth, but they are not as good as visible as the depressions of S1 and S2.

In the Zechstein Formations z2 to z3 a large LRZ is observed between 60 m and 110 m profile length at ca. 60 m to 90 m depth.

## 4.5 Geological interpretation

Combining the information of the geological map, the seismic profiles and the boreholes give the following interpretation. The subsurface below Schmalkalden has been affected by tectonic movements since at least the Mesozoic, when a NW–SE striking, dextral strike-slip fault zone containing the SFZ and the HFZ formed. The latter crosses the subsurface of the town of Schmalkalden. In the area of the sinkhole, the strike-slip fault created a zone of reverse, normal and strike-slip faults, as imaged by the seismic profiles (Fig. 10).

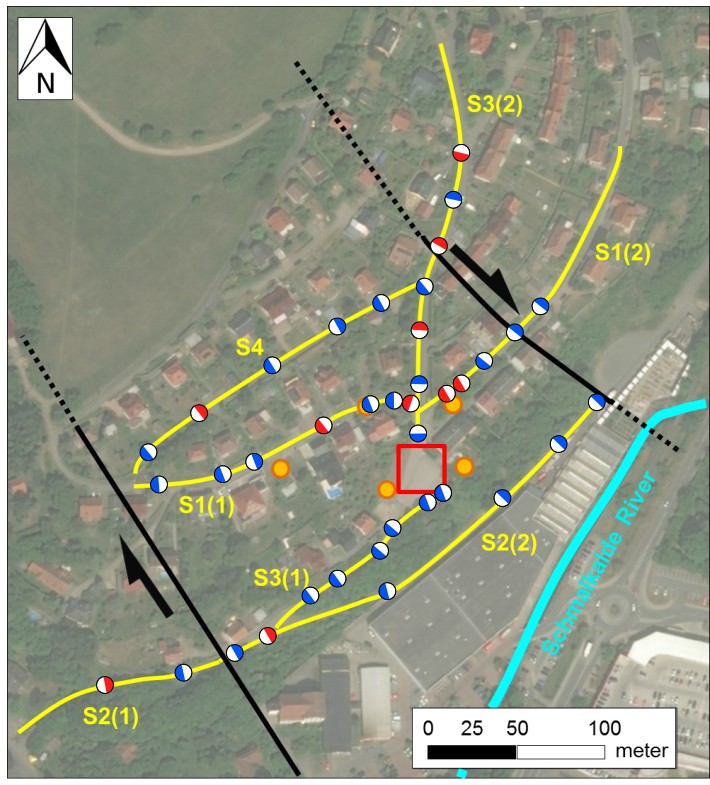

**Figure 10.** Map showing the complex fault geometry around the sinkhole. The faults identified in the $S_H$-wave seismic were extrapolated to the surface. Blue and red balls represent normal and reverse faults, respectively. The white areas within the balls display the position of the hanging wall. Note the high fault density north of the sinkhole area with fault blocks less than 10 m wide. For borehole information (orange dots), refer to Fig. 3.





We observe local thickening of the Triassic Calvörde Formation, which can only be accounted for by syn-tectonic sedimentation during the Triassic. These normal faults generated additional accommodation space for the terrestrial fluvial sediments. This was part of the multi-phase tectonic deformation which has been recorded in southern Thuringia (Wunderlich, 1997b). The extensional stress regime during the Upper Permian to the Lower Cretaceous and the compressional stress regime during the Upper Cretaceous to the Tertiary produced the dextral strike-slip zone, and the various strike-slip faults within it. The fine mosaic of fault blocks is mainly attributed to this latter phase.

### 4.6 Fault inventory

The complex 3-D structure of the faults is difficult to decipher with 2-D seismic lines, even if they are densely spaced. Nevertheless, large (5 m to 10 m) displacements of the reflectors were found at several locations along the profiles, and they are interpreted as near-surface normal and reverse faults, e.g., below the western margin of the depression structure, steep, northeast-dipping reflectors are visible in profile S1 at a depth of 50 m to 100 m. The borehole 05/2011 proved the existence of a fault in this zone, which was interpreted as a northeast-dipping normal fault.

Figure 10 shows an overview of all fault positions extrapolated to the surface. The high density of faults means that fault blocks are less than 50 m wide and sometimes less than 10 m, especially directly north of the sinkhole. In general there ís a mixture of apparent normal (25 faults) and reverse (10 faults) faults. All faults have an apparent dip angle of greater than 70° (apparent angle because they are measured in a 2-D section). This means the seismic profile is close to the true dip direction of the faults, because otherwise the faults would not be so steep. Consequently, the faults strike roughly NW–SE, similar to the strike of the HFZ (Figs. 4 and 10). There is a tendency for normal faults on the southwest end of the profiles to dip northeast and vice versa. Note also that a few faults are outside of the presumed fault zone.

Interestingly, reverse faults cannot be followed from one seismic section to another (e.g. S1 and S4, Fig. 10), meaning that either the strike length of the fault is less than the distance between seismic sections (e.g. less than 50 m), which is most unlikely, or that a reverse fault in one profile correlates with a normal fault in another profile. The significance of this is described in detail in the next section.

## 5 Discussion

The fact that steep normal and reverse faults occur side by side, as seen in the four seismic sections, could be due to two mechanisms, both related to strike-slip faults. Either some of the normal faults that were generated under extension during the deformation phase from the Upper Permian to the Lower Cretaceous, were inverted by later compression during the deformation phase from the Upper Cretaceous to the Tertiary (Wunderlich, 1997b). For instance fault bends may switch from transtensional to transpressional systems or vice versa, if the original fault bend was at a low angle relative to the maximum horizontal stress (Tikoff & Teyssier, 1994; Legg et al., 2007). Alternatively, if there are jogs along strike of a strike-slip fault, this will produce constraining and releasing bends (Cunningham & Mann, 2007), which will cause, after movement, reverse




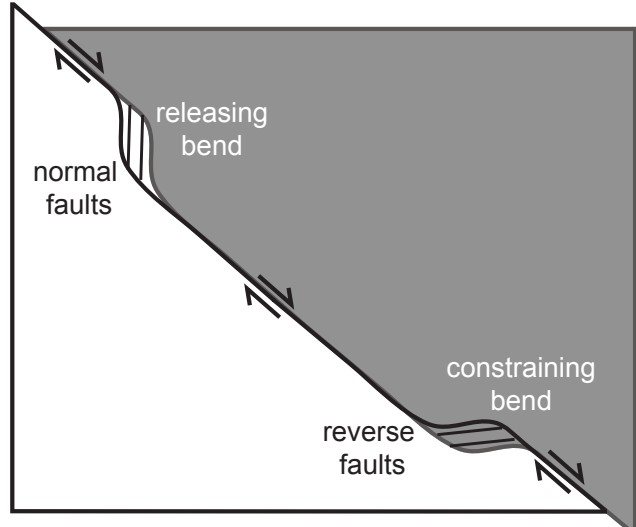

**Figure 11.** Cartoon to demonstrate how jogs in a strike-slip fault can cause both constraining and releasing bends. After right-lateral movement, normal and reverse faults are created at the releasing and constraining bends, respectively. The strike of the new faults will be close to that of the strike-slip fault, depending on the original jog angle. Effectively, it creates a system of normal and reverse faults with similar strike, along the strike of the strike-slip fault.

and normal faults, respectively, with similar strike to the strike-slip fault, along the strike of the strike-slip fault (Crowell, 1974; Christie-Blick & Biddle, 1985; Gamond, 1987, Fig. 11).

The high fault density and the complex fault geometry in the research area (Fig. 10) did not allow to make a direct spatial correlation of the faults, e.g. connecting the faults that were identified in two 2D seismic profiles. Only a high-resolution 3D

shear-wave reflection seismic survey could deliver more or less unquestionable spatial correlations, but such a technique is still in development. Nonetheless, we were able to identify the 2D fault geometries and displacements from the 2D reflection seismic profiles.

The presence of a fault or a fault zone is not the only condition that has to be fulfilled for the occurrence of a sinkhole like the one in Schmalkalden. Faults can be classified as open or sealed faults. A fault seal due to clay smear or mineralization

could hamper subrosion because it reduces fluid pathways (Caine et al., 1996; Evans et al., 1997; Ngwenya et al., 2000). On the other hand an open fault can serve as a fluid pathway.

During displacement along a fault especially the hanging wall undergoes deformation caused by the fault morphology and the resulting strain variations. The variations in strike of secondary faults are the direct result of these strain variations (Lohr, 2008). These small strike variations can be observed in Schmalkalden, but besides the faults and fractures visible in

the seismic sections subseismic scale deformation will have occurred too. Displacements along faults can result to a high fracture density around and between faults, creating a damage zone, which has the potential to increase the fluid flow due to enhanced permeability. The key factors for the increase in fracture density are the change in mechanical rock properties,



interactions between faults and a change in fault geometry (Bolas & Hermanrud, 2003; Gartrell et al., 2004; Leckenby et al., 2005; Eichhubl et al., 2009; Kim & Sanderson, 2010; Ziesch, 2016). The $S_H$-wave reflection seismic profiles carried out in this study identified a complex local and regional fault system with a dense fracture network, which enables the groundwater to circulate through the evaporites and therefore it enhances subrosion due to an increase in permeability. Different joint sets

were observed in outcrops in the vicinity of the sinkhole: (1) steep joints and fracture with a NW-SE striking, (2) flat joints and fractures with a NW-SE striking, (3) NE-SW striking joints and fractures, (4) young NNE-SSW striking joints and (5) young NNW-SSE striking joints (Schmidt et al., 2013).

The sinkhole of Schmalkalden is located at the meeting point of three groundwater catchment areas, the first is to the north and belongs to the Gespringe Spring. The second is ca. 200 m south of the sinkhole and belongs to the confluence of the Schmal-

kalden River (Fig. 4) and the Stille River. The third is to the west and belongs to the Mittelschmalkalde River. Four groundwater levels are found in the Quaternary gravel, the Lower Triassic sandstone, the Leine Carbonate (z3) and the Paleozoic bedrock. The latter consists of deep thermal, mineralized water, which is undersaturated with regard to sulfates. This is an indicator of a short residence time and a high flow rate, and the more undersaturated the water, the more sulfates can be dissolved. The main groundwater level situated in the Zechstein Formations actively leaches the soluble Permian deposits. This Zechstein water

ascends along faults and fractures and mixes with water from upper groundwater levels, since no widespread vertical separation of groundwater levels is available due to tectonics and subrosion (Henke, 1983; Schmidt, 1995). The groundwater runoff follows the morphological gradient, and at the steep faults and the intersections of faults the artesian-confined groundwater can migrate upward and leach the soluble Permian deposits. Tracer test revealed a flow rate of $100\,\mathrm{m\,h^{-1}}$ to $150\,\mathrm{m\,h^{-1}}$ (Schmidt, 1995), which is a typical value for fractured and karstic aquifer (Ravbar et al., 2012).

Groundwater table contour plans reveal a change in flow direction around the area of the HFZ (Fig. 12). To the north, the groundwater contour lines run from north to south with a flow direction from east to west and from west to east towards the Werra River. But in the area of the HFZ the contour lines run approximately east-west with a flow direction from north to south and from south to north towards the Schmalkalde River. This change in groundwater flow direction may be another reason for the occurrence of the sinkhole and can be correlated with the faults discovered in this study, because steep-dipping faults

are assumed to be barriers for horizontal groundwater flow perpendicular to the faults but serve as conduits for horizontal flow along the faults (Bredehoeft et al., 1992; Bense et al., 2003). The NW-SE striking fault branches of the HFZ hamper the groundwater flow coming from the east from the Thuringian Forest towards the Werra River; as a result the water flows from north to south along the faults towards the Schmalkalde River and thereby passes through the sinkhole area.

Several other studies regarding sinkhole distribution have shown the clustering of sinkholes along fault lineaments (Abelson

et al., 2003; Doctor et al., 2008) and the decreasing number of sinkhole occurrences with increasing distance from the fault (Hyland et al., 2006; Billi et al., 2007). A lineament of sinkholes can also be used to find hidden faults (Closson & Karaki, 2009). This is the case in Schmalkalden since the only sinkhole in the urban area formed within the strongly fractured HFZ. A single fault within the soluble rocks might not have influenced the groundwater flow direction and the upward migration of artesian-confined groundwater that much and might not have triggered a collapse due to subrosion. But since the soluble





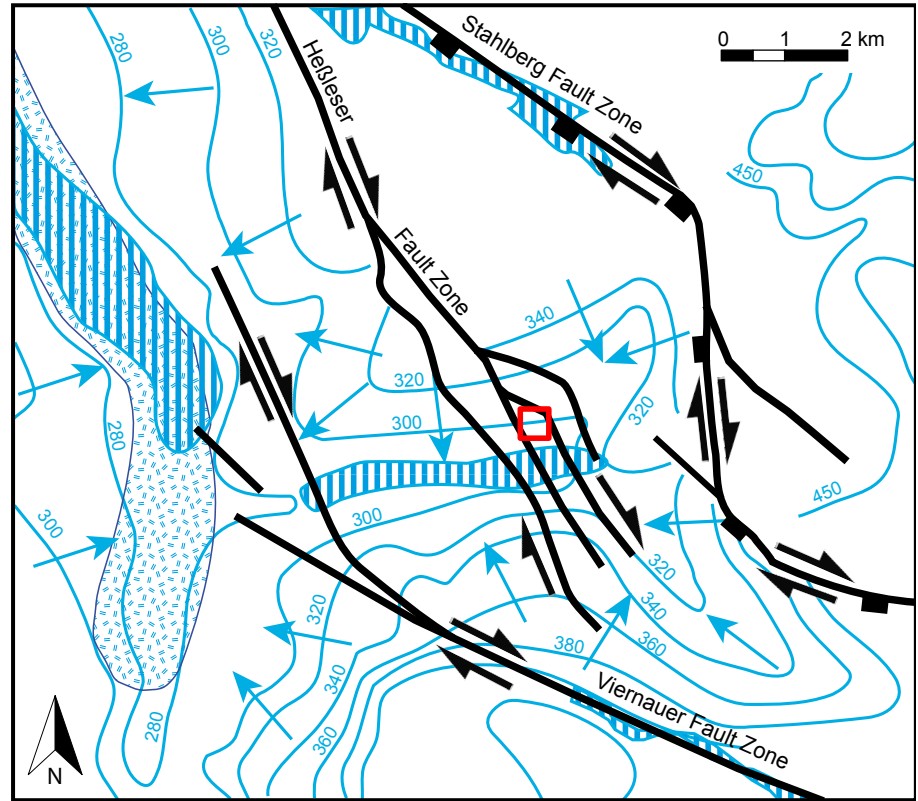

**Figure 12.** Groundwater table contour plans (after TLUG, 2017b) reveal a change in flow direction around the area of the HFZ (black lines). To the north, the groundwater contour lines (blue lines) run from north to south with a flow direction from east to west and from west to east (blue arrows). But in the area of the HFZ the contour lines run approximately east-west with a flow direction from north to south and from south to north. This change in groundwater flow can be correlated with the faults, because steep-dipping faults are assumed to be barriers for horizontal groundwater flow perpendicular to the faults but serve as conduits for horizontal flow along the faults (Bredehoeft et al., 1992; Bense et al., 2003). As a result the groundwater passes through the sinkhole area (red square). Areas with artesian-confined groundwater are marked with vertical blue lines and area with salt water ascension are marked with strokes.

rocks are located within a strike-slip fault zone with a typically strongly-fractured underground and increased permeability, as described above, the subrosion process is greatly enhanced.

We assume that elongated cavities formed along the faults planes and at fault intersections these cavities became larger and migrated upward over time. As the sustainability of the overburden was exceeded the cavity collapsed and a sinkhole formed.

5  Additionally, the subrosion-induced depressions within the Quaternary and Triassic deposits and the Zechstein evaporites that are displaced along the steep-dipping normal faults and which show local thinning, are a result of the leaching processes which occur along the faults. Possible indicators for unstable zones and subrosion processes are the LRZ's observed in the seismic data. They are visible at the depth of the subrosion horizon and below subrosion-induced features such as depressions. A LRZ




can be an indicator for an area in which the evaporites have already been leached or are being leached due to the contact with groundwater. This can also be the reason why the subrosion horizon itself does not generate a strong impedance contrast.

Subrosion creates structures at a subseismic scale which cannot be imaged because the resolution limit of the $S_H$-seismic at such depths is about 3 to 4.5 m and as a result no distinct reflectors are observed. In previous work by Krawczyk et al. (2012)
and Wadas et al. (2016) carried out in Hamburg and Bad Frankenhausen in Germany, respectively, similar subrosion-induced features were observed. The seismic sections show a lateral and vertical variable reflection pattern with discontinuous reflectors and small-scale fractures formed by subrosion of soluble rocks near the surface.

Besides reflection seismic, other geophysical investigations were carried out in Schmalkalden on behalf of TLUG, e.g., gravime­try. Gravimetry is sensitive to density and due to the fault-enhanced subrosion the density varies vertically and laterally. The
micro-gravimetric surveys show a negative anomaly with a WNW–ESE extension which crosses the sinkhole and the surround­ing area (Schmidt et al., 2013). This anomaly can be linked to the fault system which crosses the sinkhole area. The source for the gravimetric anomaly is located at a depth of 50 m to 100 m, which coincides with the subrosion horizon (Seidel & Serfling, 2010).

Shear-wave reflection seismic has proven to be a suitable method to image and analyze near-surface subrosion structures
and zones in high-resolution, not only for the case of Schmalkalden, but also in Hamburg (Krawczyk et al., 2012) and Bad Frankenhausen (Wadas et al., 2016) in Germany, and at the Dead Sea in Jordan (Polom et al., 2016b). The method is applicable to other study areas, if its limitations are kept in mind, like the inferior penetration depth, compared to P-wave refection seismic, especially in subrosion areas, and the resolution limits. We suggest to use it as a support in addition to P-wave reflection seismic to get more structural information of areas where the P-wave has insufficient resolution.

## 6 Conclusions

In this study we used $S_H$-wave reflection seismic to analyze subrosion features, and the link between faults, groundwater flow and soluble rocks.

Areas affected by tectonic deformation phases are prone to enhanced subrosion. The deformation of the fault blocks leads to the generation of a damage zone around and in between the faults with a dense fracture network, which enables the groundwater
to flow through the subsurface and to leach soluble rocks. The more complex the fault geometry and the more interaction between faults, the more fractures are generated e.g. subrosion in a strike slip-fault zone with steep normal and reverse faults will be enhanced more than in an area with just a simple normal or reverse fault.

The faults do not only serve as fluid pathways for groundwater to migrate upward, they also influence the entire groundwater flow direction. We have shown that in a sinkhole area with a broad zone of parallel fault branches, steep-dipping faults can act
as barriers for horizontal groundwater flow but instead can serve as conduits for water to flow parallel to the fault strike and will thus cause subrosion.

Structural analysis of S-wave seismics is a valuable tool for the detection of near-surface faults in order to characterize the faulting in an area and thus wether it is prone to subrosion or not. The reconstruction of strike and dip directions of even small




fault blocks can help to better understand the hydrodynamic groundwater conditions, which is another important key factor for understanding the subrosion process in general.

*Data availability.* The seismic data is the property of LIAG. The data is available from the first author upon request. Please contact Sonja H. Wadas for details.

5 *Competing interests.* The authors declare that they have no conflict of interest.

*Acknowledgements.* We thank our colleague Saskia Tschache and LIAG's seismic data acquisition and technical development team, Jan Bayerle, Eckhardt Großmann, Sven Wedig, and Erwin Wagner for their excellent work during the field surveys. Furthermore we thank Lutz Katzschmann, Sven Schmidt and Ina Pustal from the Thuringian State Institute for Environment and Geology (TLUG) who provided the borehole information and helped improve the manuscript.



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



## Appendix A:  Shear-wave reflection seismic data

Table  A1 shows detailed information about the equipment, the acquisition parameters, and geometry used for the SH-wave reflection seismic survey carried out around the sinkhole of Schmalkalden. For detailed explanations regarding the equipment for the high-resolution shear wave reflection seismics, see Polom et al. (2010); Krawczyk et al. (2012, 2013); Polom et al.

5   (2013).

The data processing here was based on general processing procedures, as described by Krawczyk et al. (2012) and Pugin et al. (2013). Table  A2 shows the processing steps applied to the SH-wave reflection seismic profiles. Most of the processing steps were carried out iteratively and were adjusted for each profile to get the best results. For detailed explanations regarding the processing algorithms see Hatton et al. (1986); Lavergne (1989); Baker (1999); Yilmaz (2001).



**Table A1.** Acquisition parameters of the SH-wave reflection seismic surveys.

| Acquisition parameters | SH-wave reflection seismic |
| --- | --- |
| Source type | Electrodynamic vibrator ELVIS 7 |
| Source signal | 20 – 120 Hz (10 sec.; linear) |
| Source spacing | 2 / 4 meters |
| Receiver type | 1C-geophones (horizontal; 10 Hz) |
| Number of receivers | 112 / 120 |
| Receiver spacing | 1 meter |
| Recording system | Geometrics Geode |
| Record length | 12 sec. uncorrelated |
| Sampling interval | 1 ms |
| Record number | 1446 |
| Profile length | total ca. 1280 m |

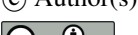



**Table A2.** General processing sequence applied to SH-wave seismic reflection data. Most of the processing was carried out iteratively and each profile is individualized due to the differing data quality of the profiles.

| Processing step | SH-wave reflection seismic |
|---|---|
| Correlation | Vibroseis-correlation of sweep/recorded signal to compress the time-stretched signal |
| Geometry | Geometry assignment |
| Amplitude balancing & frequency filtering | AGC (220 ms length), bandpass filter (16/18 Hz – 96/98 Hz) & trace energy normalization |
| Vertical stack | 2-fold |
| Zeroing of traces | Top mute for individual zeroing of amplitudes at the top of the records |
| Spectral balancing | Bandwidth range 20 Hz, slope 5 Hz, start 16 Hz, end 95 Hz, not applied to all profiles |
| 2D filter | Frequency-wavenumber (FK) filter (removal of surface waves, harmonic distortions & coherent noise) |
| Data sort | Sort to CMP gathers |
| Velocity analysis | Interactive velocity picking in using semblance, offset gathers, and constant velocity stacks (interval 5 m to 20 m) |
| Normal moveout & stack | NMO correction to shift reflection hyperbolas to zero-offset traveltimes, followed by CMP trace stack |
| Filter | Remaining noise is removed using a bandpass or FK filter |
| Migration | FD time migration using smoothed stacking velocities to move dipping reflectors to their true subsurface position and to collapse diffractions |
| Depth conversion | Conversion from time section to depth section |