# Peer review of "Structural analysis of S-wave seismics around an urban sinkhole; evidence of enhanced dissolution in a strike-slip fault zone"

_Natural Hazards and Earth System Sciences, 2017_

## Referee Comment (RC1) · Anonymous Referee #1 · 11 Sep 2017

1. General comments

When dealing with sinkholes, the fundamental questions are: 1. At which depth the cavity formed?

2. Why the cavity formed?

3. Is it an isolated phenomenon or is it widespread?

4. How much time was needed for the cavity to reach the ground surface? And consequently when did the cavity formed?

5. Is the process at the origin of the cavity persists? Or is there another cavity below

[Figure]

the one studied?

So far, this paper provides a convincing answer to the first three questions.

Based on the rich datasets (boreholes, gravimetric, seismic profiles etc.), the authors could at least discuss on the fourth question.

Regarding the fifth question - which is quite important for the validity of the comprehensive model developed in this paper... and for the safety of the inhabitants - it is unclear from the seismic profiles and boreholes if the answer is positive or negative.

Indeed, based on the submitted text and the elements explaining the formation of the cavity (tectonic blocks, convergence of unsaturated underground water, etc.), it seems that the process leading to the cavity formation will not stop because all the elements necessary for the cavity formation are still present after the (first?) sinkhole reached the surface.

It would be interesting to discuss this point in the dedicated section based on the seismic profile and also the gravimetric data.

Finally, if this sinkhole is an isolated case, would it be possible that the cavity had an anthropogenic origin (e.g. over pumping, mining activities in the surroundings)?

Globally, this paper is very well written. It is a model of geological and geophysical rigor. Technically it can be published after minor revision.

2. Specific comments

In my opinion, the section related to the geological evolution should be simplified in order to keep only the information strictly necessary for the explanations given to the five questions above mentioned.

For example, the author could draw a simplified cross section with the different lithologic layers and could underline the specific horizon where the cavity was most probably created.

The tectonic setting is an important parameter and a dedicated figure showing the different stages would help in the understanding.

The figure dedicated to the underground water circulation is fundamental for the overall understanding and should be improved / redrawn in order to:

- support the sentence in the text related to the confluence zone (page 15, line 8) "The sinkhole of Schmalkalden is located at the meeting point of three groundwater catchment areas";

- explain the different symbol used (rather than in the text only), and also the elevations;

- Improve the relationship with the tectonic setting (tectonic blocks).

The discussion section is rated long and should be summarized in order to highlight the key elements such as the water circulation, the micro-gravimetric results, etc. I suggest drawing a table to list the contributions, the results, and the facts to take into account in the explanatory model.

3. Technical corrections

Page 1, line 17: Closson, D. & Karaki -> Closson, D. & Abou Karaki, N.

Page 6, line 7: istallation -> installation

Page 15, line 31: Closson, D. & Karaki, N.A. -> Closson, D. & Abou Karaki, N.

Page 19: Closson, D. & Karaki, N.A. -> Closson, D. & Abou Karaki, N.

Nice work, congratulation!
* * *

---

## Referee Comment (RC2) · Anonymous Referee #2 · 19 Sep 2017

General comments: The paper analyses the area around a large collapse sinkhole that formed on 2010 in Schmalkaden, Germany, using several shear wave seismic reflection profiles and multiple boreholes, in order to unravel the factors that controlled the development of the subsidence phenomena. The sinkhole was related to the dissolution of Permian evaporites at a depth of 50-100 m, in an area where the Phanerozoic bedrock is affected by an inactive system of NW-SE strike slip faults. Authors infer from the profiles a dense network of steeply dipping dip-slip tectonic faults (normal and reverse). They conclude that faults contributed to increase permeability and controlled groundwater flow, favouring "subrosion" processes (dissolution and subsidence). On the one hand, this is not a relevant scientific finding. It is well known that faults may

favour sinkhole development by increasing permeability, guiding groundwater flow and reducing the mechanical strength of the rocks. On the other hand, it is not clear that the small-throw faults imaged in the seismic profiles are true tectonic faults; they are depicted with as continuous lines in the supra-evaporitic units and with dashed lines in the evaporites. They could correspond to gravitational collapse faults related interstratal dissolution of the evaporites and subsidence of the overlying formations. This process generates both normal and pseudoreverse faults; normal faults that over-steepen close the surface and tilted normal faults with the appearance of reverse faults. In fact, authors describe: (1) bowl-shaped structures (synformal structures) larger than 150 m wide and large "subrosion-induced depressions; (2) a system of steep normal fault next to the sinkhole that apparently dies out in the evaporites; (3) variations in the thickness of the evaporites attributable to dissolution. Authors indicate that "the complex 3-D structure of the faults is difficult to decipher with 2-D seismic lines"; "the high fault density and the complex fault geometry . . . did not allow to make a direct spatial correlation of the faults. . .only a high-resolution 3D shear-wave reflection seismic survey could deliver more or less unquestionable spatial correlations".

Specific comments: I strongly suggest the authors to avoid the term "subrosion", which is rarely used in the sinkhole literature. I recommend the use of dissolution and subsidence. The setting section should include information on: (1) the geomorphology of the area, including karst features (e.g. karst depressions, previous subsidence events); (2) the hydrogeological behaviour of the different units and their broader hydrogeological context. The geological map included in figure 1 is not readable (legend, use symbols in the map indicating units).

---

## Author Comment (AC1) · 21 Sep 2017

We want to thank two anonymous referees, who invested their precious time writing the reviews and therefore helped improving the manuscript. The PDF file contains the reviewer comments, the corresponding author comments, and the differences between the two manuscript versions. Any removed words are crossed out with a single line and colored red, whereas any added words are underlined with a squiggle and colored blue. Modifications of images are described in in the table.

Please also note the supplement to this comment:

[Figure]

https://www.nat-hazards-earth-syst-sci-discuss.net/nhess-2017-315/nhess-2017-315-AC1-supplement.pdf

---

## Editor Decision (ED1)

REVIEW 1: Anonymous RC1 (minor revision)

REVIEW 2: Anonymous RC2 (minor revision)

| Section | Reviewer | Comment | agreed | not agreed | comment |
|---|---|---|---|---|---|
| **Abstract** | None | None | | | |
| **1 Introduction** | Anonymous RC1: | page 1 line 17 - Closson, D. & Karaki N.A. --> Closson, D. & Abou Karaki, N. | X | | We corrected the reference. |
| **2.1 Study area-Geological evolution** | Anonymous RC1: | Section should be simplified in order to keep only the information strictly necessary for the explanations given to the five questions above mentioned. For example, the author could draw a simplified cross section with the different lithologic layers and could underline the specific horizon where the cavity was most probably created. | X | | We removed some information from the first paragraph, but we think the remaining information is necessary for the reader to understand the local geology and its relationship with the subrosion processes. We describe in more detail the deposition of the Permian evaporites, which contain the subrosion horizons; the Triassic and Quaternary formations that are part of the interpreted seismics sections are described only in a short paragraph; and the last paragraph is necessary to understand the tectonic stress regime. The different lithological layers are visualized in Fig. 3, but to underline the subrosion horizon, we added a corresponding label to Figure 3. The same was done for Figures 6,7,8 and 9. |
| | | The tectonic setting is an important parameter and a dedicated figure showing the different stages would help in the understanding. | | X | On page 4 line 2, at the end of the paragraph, which describes two of the six tectonic phases, we refer to Wunderlich (1997b), who investigated the tectonic phases and has already drawn a detailed figure to illustrate each of the six stages. To make this more obvious to the reader we moved the reference from the end to the beginning of the paragraph. |
| **2.2 Study area-Faults** | None | None | | | |
| **2.3 Study area-Sinkhole** | None | None | | | |
| **2.4 Study area-Stratigraphy** | None | None | | | |
| **3 Shear wave data** | Anonymous RC1: | page 6 line 7 - istallation --> installation | X | | We corrected the wrong spelling |
| **4.1 Results-Interpretation S1** | None | None | | | |
| **4.2 Results-Interpretation S2** | None | None | | | |
| **4.3 Results-Interpretation S3** | None | None | | | |
| **4.4 Results-Interpretation S4** | None | None | | | |
| **4.5 Geological interpretation** | None | None | | | |
| **4.6 Fault inventory** | Anonymous RC2: | It is not clear that the small-throw faults imaged in the seismic profiles are true tectonic faults, they could correspond to gravitational collapse faults | | X | Some of the faults might be gravitational collapse faults, but most of them must have a tectonic origin, because the Heßleser Fault Zone crosses the town of Schmalkalden, and in the seismic profiles we identified not only normal, but also reverse faults that were generated by the strike-slip fault, and these can not be explained by gravitational collapse. We also explained the influence of the extensional (Upper Permian to Lower Cretaceous) and compressional stress regimes (Upper Cretaceous to Tertiary), which have influenced the regional geology by e.g. generating normal faults, as proven by other studies (we refer to our explanations in sections 2.1, 2.2 (Study area) and 5 (Discussion). |
| **5 Discussion** | Anonymous RC1: | page 15 line 31 - Closson, D. & Karaki N.A. --> Closson, D. & Abou Karaki, N. | X | | We corrected the reference. |
| | | The discussion section is rated long and should be summarized in order to highlight the key elements such as the water circulation, the micro-gravimetric results, etc. I suggest drawing a table to list the contributions, the results, and the facts to take into account in the explanatory model. | X | | We changed the last paragraph of the 'Discussion' section a little and included a brief summary of the topics we have discussed. |
| **6 Conclusions** | None | None | | | |
| **References** | Anonymous RC1: | page 19 line 25 - Closson, D. & Karaki N.A. --> Closson, D. & Abou Karaki, N. | X | | We corrected the reference. |
| **Figures** | Anonymous RC1: | Figure 12: The figure dedicated to the underground water circulation is fundamental for the overall understanding and should be improved / redrawn in order to: - support the sentence in the text related to the confluence zone (page 15, line 8) "The sinkhole of Schmalkalden is located at the meeting point of three groundwater catchment areas"; explain the different symbol used (rather than in the text only), and also the elevations; Improve the relationship with the tectonic setting (tectonic blocks) | X | | We added groundwater confluence lines in order to better visualize the confluence zone at the sinkhole area and that this meeting point is located directly in the middle of the fault blocks of the HFZ. We also added a legend below the figure. |
| **Appendix** | None | None | | | |
| **General comments** | Anonymous RC1: | When dealing with sinkholes, the fundamental questions are: 1. At which depth the cavity formed? 2. Why the cavity formed? 3. Is it an isolated phenomenon or is it widespread? 4. How much time was needed for the cavity to reach the ground surface? And consequently when did the cavity formed? 5. Is the process at the origin of the cavity persists? Or is there another cavity below the one studied? So far, this paper provides a convincing answer to the first three questions. Based on the rich datasets (boreholes, gravimetric, seismic profiles etc.), the authors could at least discuss on the fourth question. Regarding the fifth question - which is quite important for the validity of the comprehensive model developed in this paper and for the safety of the inhabitants - it is unclear from the seismic profiles and boreholes if the answer is positive or negative. | X | | We added two paragraphs in the 'Discussion' section in which we discuss the fourth and fifth questions regarding the time needed for sinkhole developement and the still ongoing subrosion process. We also inserted the answers to the questions in the 'Conclusions' and the 'Abstract' sections. |
| | | if this sinkhole is an isolated case, would it be possible that the cavity had an anthropogenic origin (e.g. over pumping, mining activities in the surroundings)? | X | | On behalf of TLUG, invetisgations were conducted to determine whether the sinkhole might be of anthropogenic origin, but they did not find a large underground facility or man-made cavity that could generate such a large sinkhole. We added a paragraph at the end of section 2.3 in order to explain this issue. |
| | Anonymous RC2: | I strongly suggest the authors to avoid the term "subrosion", which is rarely used in the sinkhole literature. I recommend the use of dissolution and subsidence. | | X | The term 'subrosion' has been used in older studies e.g. 'Salt Dissolution in Subsurface of British North Sea as Interpreted from Seismograms' by Hans Lohmann (1972); 'Subsidence and related features in the Tully Valley, central New York' by Getchell (1996); Syndepositional and postdepositional salt-dissolution structure traps in Lower Cretaceous sandstones by Hopkins and Pollock (1988), but is also widely used in studies and books of recent years, as shown by a Scopus document search (database of all Elsevier Journals and others) of the term 'subrosion'. |
| | | The setting section should include information on: (1) the geomorphology of the area, including karst features (e.g. karst depressions, previous subsidence events); (2) the hydrogeological behaviour of the different units and their broader hydrogeological context. | X | | We added information about the occurence of saltwater springs in section 2.3. Information on the hydrological behaviour and the different groundwater levels are given in the discussion section (see pages 15 and 16 of revised manuscript). |
| | | The geological map included in figure 1 is not readable (legend, use symbols in the map indicating units). | X | | We changed the legend and the map for better readability. |

Lohmann, H.L.: Salt Dissolution in Subsurface of British North Sea as Interpreted from Seismograms. AAPG Bulletin , 56, (3), 472-479, 1972. Citation from Abstract: 'Salt removal from a certain place in the subsurface can result from either halokinetic salt flow or leaching (= subrosion). '

Getchell, F.J.: Subsidence and related features in the Tully Valley, central New York. International Journal of Rock Mechanics and Mining Sciences and Geomechanics, 33, (2), 1996.

Hopkins, J.C. & Pollock, J.E.: Syndepositional and postdepositional salt-dissolution structure traps in Lower Cretaceous sandstones, Cessford Field, Alberta, Canada. AAPG Annual Meeting 1988, url =https://www.osti.gov/scitech/biblio/6776250. Citation from Abstract: 'Many Cretaceous structure-stratigraphic traps in the Western Canada sedimentary basin formed through regional tilting and/or local salt dissolution (subrosion).'

[revised manuscript text omitted]

**Figure 12.** Groundwater table contour  map (after TLUG, 2017b)  reveals a  confluence zone of three different groundwater bodies.  Three groundwater  bodies show different flow  directions towards the sinkhole area, and  the change in groundwater flow can be  see to correlate with the faults~~, because steep-dipping faults are assumed to be barriers for horizontal groundwater flow perpendicular to the faults but serve as conduits for horizontal flow along the faults (Bredehoeft et al., 1992; Bense et al., 2003).As a result the groundwater passes through the sinkhole area (red square). Areas with artesian-confined groundwater are marked with vertical blue lines and area with salt water ascension are marked with strokes.~~

ing area (Schmidt et al., 2013). This anomaly can be linked to the fault system which crosses the sinkhole area. The source for the gravimetric anomaly is located at a depth of 50 m to 100 m, which coincides with the  horizon (Seidel & Serfling, 2010).

 We assume that the  processes are still ongoing, because the key factors that lead to

5    leaching of soluble rocks are still present, e.g. the complex strike-slip fault system and the resulting fractured subsurface with

its fluid pathways, the soluble rocks, and the confluence zone of three groundwater catchment areas. In the seismic section, several depressions were identified that might develop into sinkholes in the future, if the cavities reach the surface, e.g. the large depression northwest of the sinkhole, which also coincides with the WNW–ESE trending negative gravimetry anomaly.

In summary, we discuss the discovery of steep, normal and reverse faults in the seismic sections, which are part of a strike-slip fault zone. The faults are positioned very closely, forming a dense fracture network, not only on the seismic scale, but also scale. They act as fluid pathways and lead to fault-enhanced subrosion. We describe that a LRZ is a possible indicator for an unstable zone and we show that gravimetry measurements our results.

Altogether, shear-wave reflection seismic measured at Schmalkalden has proven to be a suitable method to image and analyze near-surface subrosion structures in high-resolution. This is also the case in Hamburg (Krawczyk et al., 2012), Bad Frankenhausen (Wadas et al., 2016) in Germany, and at the Dead Sea in Jordan (Polom et al., 2016b). The method is applicable to other study areas, if its limitations are kept in mind, such as the penetration depth, with respect to P-wave refection seismic, especially in subrosion areas, and the limits of resolution. We suggest it should be used to support P-wave reflection seismics, to get more structural information in areas where the P-wave has insufficient resolution.

**6 Conclusions**

In this study we used $S_H$-wave reflection seismic to analyze subrosion features, and the link between faults, groundwater flow and soluble rocks.

Areas affected by tectonic deformation phases are prone to enhanced subrosion. The deformation of the fault blocks leads to the generation of a damage zone around and in between the faults with a dense fracture network, which enables the groundwater to flow through the subsurface and to leach soluble rocks. The more complex the fault geometry and the more interaction between faults, the more fractures are generated, e.g. subrosion in a strike slip-fault zone with steep normal and reverse faults will be enhanced more than in an area with just a simple normal or reverse fault.

The faults do not only serve as fluid pathways for groundwater to migrate upward, they also influence the entire groundwater flow direction. We have shown that in a sinkhole area with a broad zone of parallel fault branches, steep-dipping faults can act as barriers for horizontal groundwater flow, but instead can serve as conduits for water to flow parallel to the fault strike. In Schmalkalden, the subrosion horizon in which the cavities are formed is located between the Permian Leine- and Werra Formations at ca. 70 m to 120 m depth. The long lasting subrosion processes with probably slow subrosion rates, causes widespread subrosion. The processes are still ongoing and may in the area.

[revised manuscript text omitted]